# PLAYING REPEATED GAMES WITH LARGE LANGUAGE MODELS

## ABSTRACT

Large Language Models (LLMs) are transforming society and permeating into diverse applications. As a result, LLMs will frequently interact with us and other agents. It is, therefore, of great societal value to understand how LLMs behave in interactive social settings. We propose to use behavioral game theory to study LLM's cooperation and coordination behavior. We let different LLMs play finitely repeated games with each other and with other, human-like strategies. Our results show that LLMs generally perform well in such tasks and also uncover persistent behavioral signatures. In a large set of two-player, two-strategy games, we find that LLMs are particularly good at games where valuing their own self-interest pays off, like the iterated Prisoner's Dilemma family. However, they behave sub-optimally in games that require coordination, like the Battle of the Sexes. We, therefore, further focus on two games from these distinct families. In the canonical iterated Prisoner's Dilemma, we find that GPT-4 acts particularly unforgivingly, always defecting after another agent has defected only once. In the Battle of the Sexes, we find that GPT-4 cannot match the behavior of the simple convention to alternate between options. We verify that these behavioral signatures are stable across robustness checks. Finally, we show how GPT-4's behavior can be modified by providing further information about the other player as well as by asking it to predict the other player's actions before making a choice. These results enrich our understanding of LLM's social behavior and pave the way for a behavioral game theory for machines.

## 1 INTRODUCTION

Large Language Models (LLMs) are deep learning models with billions of parameters trained on huge corpora of text (Brants et al., 2007; Devlin et al., 2018; Radford et al., 2018). While they can generate text that human evaluators struggle to distinguish from text written by other humans (Brown et al., 2020), they have also shown other, emerging abilities (Wei et al., 2022a). They can, for example, solve analogical reasoning tasks (Webb et al., 2022), program web applications (Chen et al., 2021), use tools to solve multiple tasks (Bubeck et al., 2023), or adapt their strategies purely in-context (Coda-Forno et al., 2023a). Because of these abilities and their increasing popularity, LLMs are on the cusp of transforming our daily lives as they permeate into many applications (Bommasani et al., 2021). This means that LLMs will interact with us and other agents –LLMs or otherwise– frequently and repeatedly. How do LLMs behave in these repeated social interactions?

Measuring how people behave in repeated interactions, for example, how they cooperate (Fudenberg et al., 2012) and coordinate (Mailath & Morris, 2004), is the subject of a sub-field of behavioral economics called behavioral game theory (Camerer, 2011). While traditional game theory assumes that people's strategic decisions are rational, selfish, and focused on utility maximization (Fudenberg & Tirole, 1991; Von Neumann & Morgenstern, 1944), behavioral game theory has shown that human agents deviate from these principles and, therefore, examines how their decisions are shaped by social preferences, social utility and other psychological factors (Camerer, 1997). Thus, behavioral game theory lends itself well to studying the repeated interactions of diverse agents (Henrich et al., 2001; Rousseau et al., 1998), including artificial agents (Johnson & Obradovich, 2022).

In this paper, we analyze the LLMs' behavioral patterns by letting them play finitely repeated games with full information and against other LLMs as well as simple, human-like strategies. The finitely

repeated games have been engineered to understand how agents should and do behave in interactions over many iterations. We focus on two-player games with two discrete actions, i.e. $2 \times 2$-games.

Analyzing their performance across families of games, we find that they perform well in games that value pure self-interest, especially those from the Prisoner's Dilemma family. However, they underperform in games that involve coordination. Based on this finding, we further focus on games taken from these families and, in particular, on the currently largest LLM: GPT-4 (OpenAI, 2023). In the canonical Prisoner's Dilemma, which assesses how agents cooperate and defect, we find that GPT-4 retaliates repeatedly, even after only having experienced one defection. Because this can indeed be the equilibrium individual-level strategy, GPT-4 is good at these games because it is particularly unforgiving and selfish. In the Battle of the Sexes, which assesses how agents trade-off between their own and their partners' preferences, we however find that GPT-4 does not manage to coordinate with simple, human-like agents, that alternate between options over trials. Thus, GPT-4 is bad at these games because it is uncoordinated. We also verify that these behaviors are not due to an inability to predict the other player's actions, and persist across several robustness checks and changes to the payoff matrices. Finally, we point to two ways in which these behaviors can be changed. GPT-4 can be made to act more forgivingly by pointing out that the other player can make mistakes. Moreover, GPT-4 gets better at coordinating with the other player when it is first asked to predict their actions before choosing an action itself.

## 2 RELATED WORK

As algorithms become increasingly more able and their decision making processes impenetrable, the behavioral sciences offer new tools to make inferences just from behavioral observations (Rahwan et al., 2022; Schulz & Dayan, 2020). Behavioral tasks have, therefore, been used in several benchmarks (Bommasani et al., 2021; Kojima et al., 2022).

Whether and how algorithms can make inferences about other agents, machines and otherwise, is one stream of research that borrows heavily from the behavioral sciences (Rabinowitz et al., 2018; Cuzzolin et al., 2020; Alon et al., 2022). Of particular interest to the social interactions most LLMs are embedded in is an ability to reason about the beliefs, desires, and intentions of other agents, or a so-called theory of mind (ToM) (Frith & Frith, 2005). Theory of mind underlies a wide range of interactive phenomena, from benevolent teaching (Vélez & Gweon, 2021) to malevolent deception (Lissek et al., 2008; Alon et al., 2022), and is thought to be the key to many social phenomena in human interactions (Hula et al., 2015; Ho et al., 2022).

Whether LLMs possess a theory of mind has been debated. For example, Kosinski (2023) argued that GPT-3.5 performs well on a number of canonical ToM tasks. Others have contested this view, arguing that such good performance is merely a function of the specific prompts (Ullman, 2023; Le et al., 2019). Yet other research has shown that chain-of-thought reasoning significantly improves LLMs' ToM ability (Moghaddam & Honey, 2023). Moreover, it has been argued that the currently largest LLM, GPT-4, manages to perform well in ToM tasks, including in the variants in which GPT-3.5 previously struggled (Bubeck et al., 2023). Thus, GPT-4's behavior will be of particular interest in our experiments.

Games taken from game theory present an ideal testbed to investigate interactive behavior in a controlled environment and LLM's behavior has been probed in such tasks (Chan et al., 2023). For example, Horton (2023) let GPT-3 participate in the dictator game, and Aher et al. (2022) used the same approach for the ultimatum game. Both show how the models' behavior is malleable to different prompts, for example making them more or less self-interested. However, all these games rely on single-shot interactions over fewer games and do not use iterated games.

Our study builds upon recent advancements in the field, which have shifted the focus from solely assessing the performance of LLMs to comparing them with human behaviors. Previous research efforts have explored various approaches to analyze LLMs, such as employing cognitive psychology tools (Binz & Schulz, 2023; Dasgupta et al., 2022) and even adopting a computational psychiatry perspective (Coda-Forno et al., 2023b).

Finally, the theory behind interacting agents is important for many machine learning applications in general (Crandall & Goodrich, 2011), and in particular, in adversarial settings (Goodfellow et al., 2020), where one agent tries to trick the other agent into thinking that a generated output is good.

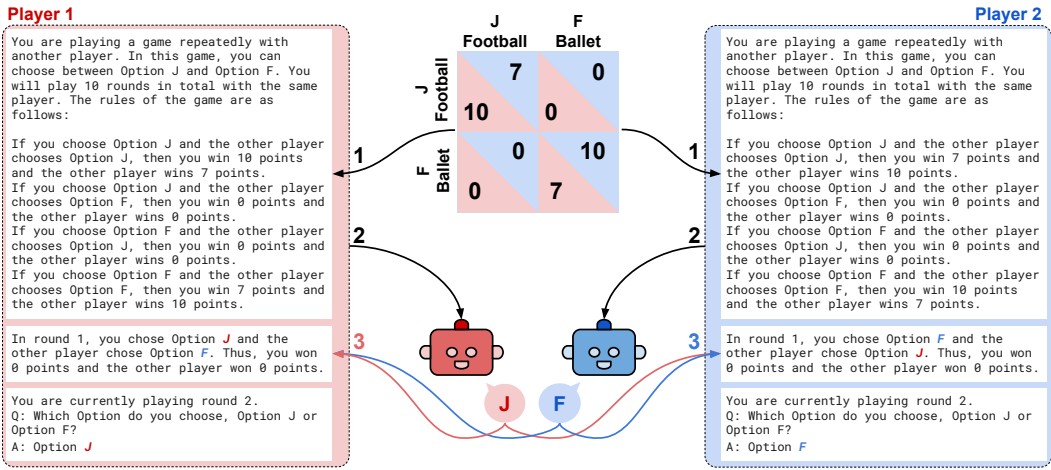

Figure 1: Playing repeated games in an example game of Battle of the Sexes. In Step (1), we turn the payoff matrix into textual game rules. (2) The game rules, the current game history, and the query are concatenated and passed to LLMs as prompts. (3) In each round, the history for each player is updated with the answers and scores of both players. Steps 2 and 3 are repeated for 10 rounds.

## 3 REPEATED GAMES METHODOLOGY

We study LLMs' behaviour in finitely repeated games with full information taken from the economics literature. We focus on two-player games with discrete choices between two options to simplify the analyses of emergent behaviors. We let two LLMs interact via prompt-chaining (see Figure 1 for an overview), i.e. all integration of evidence and learning about past interactions happens as in-context learning (Brown et al., 2020; Liu et al., 2023). The games are submitted to LLMs as prompts in which the respective game, including the choice options, is described. At the same time, we submit the same game as a prompt to another LLM. We obtain generated tokens $t$ from both LLMs by sampling from

$$p_{\text{LLM}}(\boldsymbol{t}|\boldsymbol{c}^{(p)}) = \prod_{k=1}^{K} p_{\text{LLM}}(t_k|c_1^{(p)},\dots,c_n^{(p)}, t_1,\dots,t_{k-1}) \quad (1)$$

After feeding the prompt to the LLM, our methodology is as follows. The LLM prediction of the first token following the context is $d = p_{\text{LLM}}(t_1|\boldsymbol{c}^{(p)})$ and the $N$ tokens for the possible answers of the multiple choice question are $o = \{o_i\}_{i=1}^{N}$ which in this case are J and F. The predicted option is then given by

$$\hat{o} = \arg\max(\hat{c}_i), \text{ with } \hat{c}_i = d[c_i], i = 1...N \quad (2)$$

which are the predicted probabilities of the language model. Once both LLMs have made their choices, which we track as a completion of the given text, we update the prompts with the history of past interactions as concatenated text and then submit the new prompt to both models for the next round. These interactions continue for 10 rounds in total for every game. In a single round $\pi_i(x_1, x_2)$ is the payoff for Player 1 when $x_1$ and $x_2$ are the strategies chosen by both players. In repeated games, the payoffs are often considered as discounted sums of the payoffs in each game stage, using a discount factor $\delta$. If the game is repeated $n$ times, the payoff $U_i$ for player $i$ is

$$U_i = \pi_i(x_{10}, x_{20}) + \delta \cdot \pi_i(x_{11}, x_{21}) + \delta^2 \cdot \pi_i(x_{12}, x_{22}) + \dots + \delta^{n-1} \cdot \pi_i(x_{1(n-1)}, x_{2(n-1)}) \quad (3)$$

Each term represents the discounted payoff at each stage of the repeated game, from the first game ($t = 0$) to the $n^{th}$ game ($t = n - 1$). In our experiments we keep $\delta = 1$. To avoid influences of the particular framing of the scenarios, we only provide barebones descriptions of the payoff matrices (see example in Figure 1). To avoid contamination through particular choice names or the used framing, we use the neutral options 'F' and 'J' throughout (Binz & Schulz, 2023).

**Games Considered.** We first investigate 144 different $2 \times 2$-games where each player has two options, and their individual reward is a function of their joint decision. For two additional games, Prisoner's Dilemma and Battle of the Sexes, we also let LLMs play against simple, hand-coded strategies to understand their behavior in more detail.

**Large Language Models Considered.** In this work, we evaluate five LLMs. For all of our tasks, we used the public OpenAI API with the GPT-4, `text-davinci-003` and `text-davinci-002` models which are available via the completions endpoint, Meta AI's Llama 2 70B chat model which has 70 billion parameters and is optimized for dialogue use cases, and the Anthropic API model Claude 2 to run our simulations. Experiments with other popular open source models MosaicPre-trainedTransformer (MPT), Falcon and different versions of Llama 2 (namely `MPT-7B`, `MPT-30B`, `Falcon-7b`, `Falcon-40b`, `Llama 2 7B`, `Llama 2 13B`) have revealed that these models did not perform well at the given tasks, choosing the first presented option more than 95% of the time independent of which option this is. Therefore, we chose not to include them in our main experiments. For all models, we set the temperature parameters to 0 and only ask for one token answer to indicate which option an agent would like to choose. All other parameters are kept as default values.

**Playing 6 Families of $2 \times 2$-Games Task Design.** While $2 \times 2$-games games can appear simple, they present some of the most powerful ways to probe diverse sets of interactions, from pure competition to mixed motives and cooperation - which can further be classified into canonical subfamilies outlined elegantly by Robinson & Goforth (2005). Here, to cover the wide range of possible interactions, we study the behaviors of GPT-4, text-davinci-003, text-davinci-002, Claude 2 and Llama 2 across these canonical families. We let all five engines play all variants of games from within the six families.

**Cooperation and Coordination Task Design.** We then analyze two games, Prisoner's Dilemma and Battle of the Sexes, in more detail because they represent interesting edge cases where the LLMs performed exceptionally well, and relatively poorly. We particularly hone in on GPT-4's behavior because of recent debates around its ability for theory of mind, that is whether it is able to hold beliefs about other agents' intentions and goals, a crucial ability to successfully navigate repeated interactions (Bubeck et al., 2023; Kosinski, 2023). For the two additional games, we also let LLMs play against simple, hand-coded strategies to further understand their behavior. These simple strategies are designed to assess how LLMs behave when playing with more human-like players.

## 4 EXPERIMENTS

Using GPT-4, text-davinci-002, text-davinci-003, Claude 2 and Llama 2 70B, we evaluate all types of $2 \times 2$-games. games. For the analysis of two particular games we let both all the LLMs and human-like strategies play against each other. We focus on LLMs' behavior in cooperation and coordination games.

### 4.1 ANALYSING BEHAVIOR ACROSS FAMILIES OF GAMES

We start out our experiments by letting the three LLMs play games from different families with each other. We focus on all known types of $2 \times 2$-games from the families of win-win, biased, second-best, cyclic, and unfair games as well as all games from the Prisoner's Dilemma family (Owen, 2013; Robinson & Goforth, 2005). A win-win game is a special case of a non-zero-sum game that produces a mutually beneficial outcome for both players provided that they choose their corresponding best option. Briefly, in games from the Prisoner's Dilemma family, two agents can choose to work together, i.e. cooperate, for average mutual benefit, or betray each other, i.e. defect, for their own benefit. In an unfair game, one player can always win when they play properly. In cyclic games, players can cycle through patterns of choices. Biased games are games where agents get higher points for choosing the same option but where the preferred option differs between the two players. One form of a biased game is the Battle of the Sexes, where players need to coordinate to choose the same option. Finally, second-best games are games where both agents fare better if they jointly choose the option that has the second-best utility. We show example payoff matrices for each type of game in Figure 2.

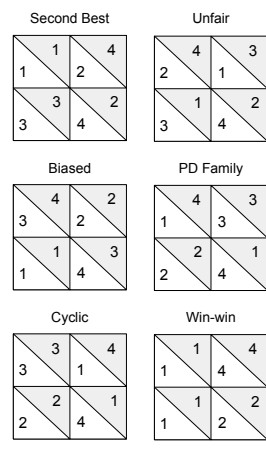

Figure 2: Canonical forms of payoff matrices for each game family.

Table 1: Performance of all models on 6 families of $2 \times 2$-games. Model score divided by maximum score achievable under ideal conditions.

| Game family | Llama 2 | Claude 2 | davinci-002 | davinci-003 | GPT-4 |
|---|---|---|---|---|---|
| Second best | 0.486 | 0.735 | 0.473 | 0.692 | **0.763** |
| Biased | 0.632 | 0.794 | 0.629 | 0.761 | **0.798** |
| Cyclic | 0.634 | 0.749 | 0.638 | 0.793 | **0.806** |
| Unfair | 0.641 | 0.812 | 0.683 | 0.833 | **0.836** |
| PD Family | 0.731 | 0.838 | 0.807 | 0.841 | **0.871** |
| Win-win | 0.915 | 0.878 | 0.988 | 0.972 | **0.992** |
| Overall | 0.697 | 0.814 | 0.730 | 0.839 | **0.854** |

We let all engines play with every other engine, including themselves, for all games repeatedly over 10 rounds and with all engines as either Player 1 or Player 2. This leads to 1224 games in total: 324 win-win, 63 Prisoner's Dilemma, 171 unfair, 162 cyclic, 396 biased, and 108 second-best games.

To analyze the different engines' performance, we calculated, for each game, their achieved score divided by the total score that could have been achieved under ideal conditions, i.e. if both players had played such that the player we are analyzing would have gained the maximum possible outcomes on every round. The results of this simulation are shown across all game types in Table 1. We can see that all engines perform reasonably well. Moreover, we can observe that larger LLMs generally outperform smaller LLMs and that GPT-4 performs best overall.

We can use these results to take a glimpse at the different LLM's strengths. That LLMs are generally performing best in win-win games is not surprising, given that there is always an obvious best choice in such games. What is, however, surprising is that they also perform well in the Prisoner's Dilemma family of games, which is known to be challenging for human players (Jones, 2008). We can also use these results to look at the different LLM's weaknesses. Seemingly, all of the LLMs perform poorly in situations in which what is the best choice is not aligned with their own preferences. Because humans commonly solve such games via the formation of conventions, we will look at a canonical game of convention formation, the Battle of the Sexes, in more detail later.

## 4.2 COOPERATION AND COORDINATION GAMES

In this section, we analyze the interesting edge cases where the LLMs performed well, and relatively poorly in Section 4.1. To do so, we take a detailed look at LLMs' behavior in the canonical Prisoner's Dilemma and the Battle of the Sexes next.

### 4.2.1 PRISONER'S DILEMMA

We have seen that LLMs perform well in games that contain elements of competition and defection. In these games, a player can cooperate with or betray their partner. When played over multiple interactions, these games are an ideal test bed to assess how LLMs retaliate after bad interactions.

In the canonical Prisoner's Dilemma, two agents can choose to work together, i.e. cooperate, for average mutual benefit, or betray each other, i.e. defect, for their own benefit and safety. In our payoff matrix, we adhere to the general condition of a Prisoner's Dilemma game in which the payoff relationships dictate that mutual cooperation is greater than mutual defection whereas defection remains the dominant strategy for both players: Crucially, the set-up of the game is such that a rationally acting agent would always prefer to defect in the single-shot version of the game as well as in our case of finitely iterated games with knowledge of the number of trials, despite the promise of

| PD Payoff | Cooperate | Defect |
|---|---|---|
| Cooperate | (8, 8) | (0, 10) |
| Defect | (10, 0) | (5, 5) |

theoretically joint higher payoffs when cooperating. This is because Player 1 always runs the risk that Player 2 defects, leading to catastrophic losses for Player 1 but better outcomes for Player 2. When the game is played infinitely, however, or with an unknown number of trials, agents can theoretically profit by employing more dynamic, semi-cooperative strategies (Axelrod & Hamilton, 1981).

As before, we let GPT-4, text-davinci-003, text-davinci-002, Claude 2 and Llama 2 play against each other. Additionally, we introduce three simplistic strategies. Two of these strategies are simple

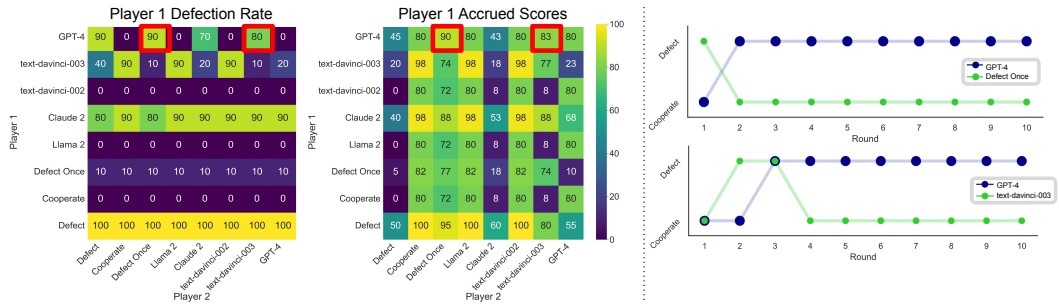

Figure 3: Overview of the Prisoner's Dilemma **Left:** Heatmaps showing Player 1 defection rate in each combination of players and scores accrued by Player 1 in each game. **Right:** Example gameplays between GPT-4 and an agent that defects once and then cooperates, and between GPT-4 and text-davinci-003. These games are also highlighted in red in the heatmaps.

singleton players, who either always cooperate or defect. Finally, we also introduce an agent who defects in the first round but cooperates in all of the following rounds. We introduced this agent to assess if the different LLMs would start cooperating with this agent again, signaling the potential building of trust.

Figure 3 shows the results of all pairwise interactions. GPT-4 plays generally well against all other agents. Crucially, GPT-4 never cooperates again when playing with an agent that defects once but then cooperates on every round thereafter. Thus, GPT-4 seems to be rather unforgiving in this setup. Its strength in these families of games thus seems to generally stem from the fact that it does not cooperate with agents but mostly just chooses to defect, especially after the other agent defected once.

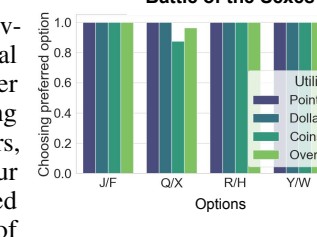

**Robustness Checks.** To make sure that the observed unforgivingness was not due to the particular prompt used, we run several versions of the game as robustness checks, modifying the order of the presented options, relabeling the options, and changing the presented utilities to be represented by either points, dollars, or coins (See Figure 4). We also added explicit end goals to our prompt, ran games with longer playing horizons and described numerical outcomes with text (See Appendix). The results of these simulations showed that the reluctance to forgive was not due to any particular characteristics of the prompts. A crucial question was if GPT-4 did not understand that the other agent wanted to cooperate again or if it could understand the pattern but just did not act accordingly. We, therefore, run another version of the game, where we told GPT-4 explicitly that the other agent would defect once but otherwise cooperate. This resulted in GPT-4 choosing to defect throughout all rounds, thereby maximizing its own points.

Figure 4: Comparing randomly presented two letter choice options and utility variants.

**Prompting Techniques to Improve Observed Behavior** One problem of these investigations in the Prisoner's Dilemma is that defecting can under specific circumstances be seen as the optimal, utility-maximizing, and equilibrium option even in a repeated version, especially if one knows that the other player will always choose to cooperate and when the number of interactions is known. Thus, we run more simulations to assess if there could be a scenario in which GPT-4 starts to forgive and cooperates again, maximizing the joint benefit instead of its own.

We took inspiration from the literature on human forgiveness in the Prisoner's Dilemma and implemented a version of the task in the vein of Fudenberg et al. (2012). Specifically, Fudenberg et al. (2012) showed that telling participants that other players sometimes make mistakes, makes people more likely to forgive and cooperate again after another player's defection (albeit in infinitely played games). Indeed, this can be favorable to them in terms of pay-offs. We observed similar behavior in GPT-4 as it started cooperating again.

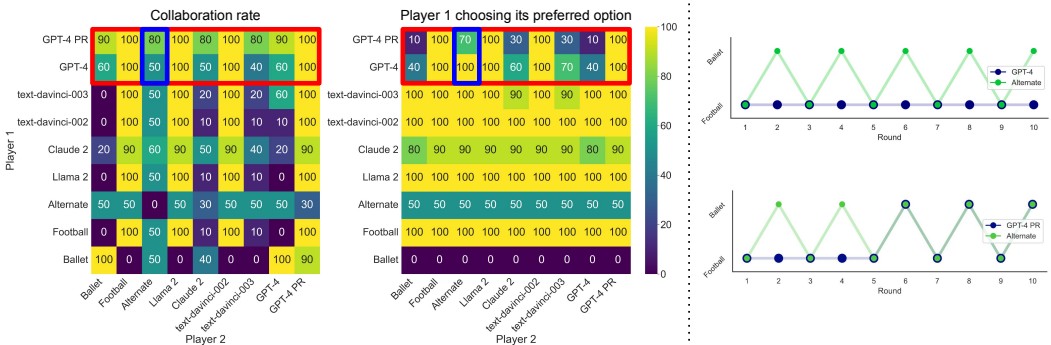

Figure 5: Overview of the Battle of the Sexes. **Left:** Heatmaps showing rates of successful collaboration between the two players and the rates of Player 1 choosing its preferred option Football. GPT-4 PR and GPT-4 performance comparisons are highlighted in **red**. **Right:** Gameplay between GPT-4 and an agent that alternates between the two options (**top**) and gameplay between GPT-4 and GPT-4 PR which represents a GPT-4 model that Predicts the opponent's move before making its own move by Reasoning about its prediction (**bottom**). Both games are also highlighted in **blue** in the heatmaps.

### 4.2.2 BATTLE OF THE SEXES

In our large-scale analysis, we saw that the different LLMs did not perform well in games that required coordination between different players. In humans, it has frequently been found that coordination problems can be solved by the formation of conventions (Hawkins & Goldstone, 2016; Young, 1996).

A coordination game is a type of simultaneous game in which a player will earn a higher payoff when they select the same course of action as another player. Usually, these games do not contain a pure conflict, i.e. completely opposing interests, but may contain slightly diverging rewards. Coordination games can often be solved via multiple pure strategies, or mixed, Nash equilibria in which players choose (randomly) matching strategies. Here, to probe how LLMs balance coordination and self-interest, we look at a coordination game that contains conflicting interests.

We study a game that is archaically referred to as the "Battle of the Sexes", a game from the family of biased games. Assume that a couple wants to decide what to do together. Both will increase their utility by spending time together. However, while the wife might prefer to watch a football game, the husband might prefer to go to the ballet. Since the couple wants to spend time together, they will derive no utility by doing an activity separately. If they go to the ballet together, or to a football game, one person will derive some utility by being with the other person but will derive less utility from the activity itself than the other person. The corresponding payoff matrix is As before, the playing agents are all three versions of GPT, Claude 2, Llama 2 as well as three more simplistic strategies. For the simplistic strategies, we implemented two agents who always choose just one option. Because LLMs most often interact with

| BoS Payoff | Football | Ballet |
|---|---|---|
| Football | (10, 7) | (0, 0) |
| Ballet | (0, 0) | (7, 10) |

humans, we additionally implemented a strategy that mirrored a common pattern exhibited by human players in the battle of the sexes. Specifically, humans have been shown to often converge to turn-taking behavior in the Battle of the Sexes (Andalman & Kemp, 2004; Lau & Mui, 2008; McKelvey & Palfrey, 2001; Arifovic & Ledyard, 2018); this means that players alternate between jointly picking the better option for one player and picking the option for the other player. While not a straightforward equilibrium, this behavior has been shown to offer an efficient solution to the coordination problem involved and to lead to high joint welfare (Lau & Mui, 2008).

Figure 5 shows the results of all interactions. While GPT-4 plays well against other agents who choose only one option, such as an agent always choosing Football, it does not play well with agents who frequently choose their non-preferred option. For example, when playing against text-davinci-003, which tends to frequently choose its own preferred option, GPT-4 chooses its own preferred option repeatedly but also occasionally gives in and chooses the other option. Crucially, GPT-4 performs poorly when playing with an alternating pattern (where, for courtesy, we let agents start with the option that the other player preferred). This is because GPT-4 seemingly does not adjust its choices to

the other player but instead keeps choosing its preferred option. GPT-4, therefore, fails to coordinate with a simple, human-like agent, an instance of a behavioral flaw.

**Robustness Checks**    To make sure that this observed behavioral flaw was not due to the particular prompt used, we also re-run several versions of the game, where we modified the order of the presented options, relabeled the options to be either numerical or other letters, and changed the presented utilities to be represented by either points, dollars, or coins as shown in Figure 4. The results of these simulations showed that the inability to alternate was not due to any particular characteristics of the used prompts. To make sure that the observed behavioral flaw was not due to the particular payoff matrix used, we also re-run several versions of the game, where we modified the payoff matrix gradually from preferring Football to preferring Ballet (or, in our case, the abstract F and J). The results of these simulations showed that GPT-4 did not alternate for any of these games but simply changed its constant response to the option that it preferred for any particular game. Thus, the inability to alternate was not due to the particular payoff matrix we used (see Appendix).

**Prediction Scenarios.**    Despite these robustness checks, another crucial question remains: Does GPT-4 simply not understand the alternating pattern or can it understand the pattern but is unable to act accordingly? To answer this question, we run two additional simulations. In the first simulation, GPT-4 was again framed as a player in the game itself. However, we now additionally ask it to predict the other player's next move according to previous rounds. In this simulation, GPT-4 started predicting the alternating pattern correctly from round 5 onward (we show this in Figure 6).

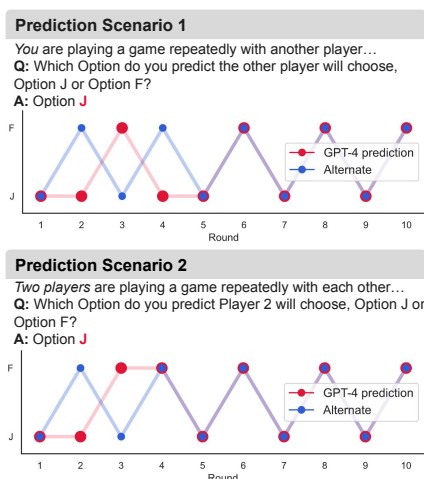

In the second simulation, instead of having GPT-4 be framed as a player itself, we simply prompted it with a game between two ('external') players and asked it to predict one player's next move according to the previous rounds. For the shown history, we used the interaction between GPT-4 and the alternating strategy. In this simulation, GPT-4 started predicting the alternating pattern correctly even earlier, from round 3 onward. Thus, GPT-4 seemingly *could* predict the alternating patterns but instead just did not act in accordance with the resulting convention. Similar divergences in abilities between social and non-social representations of the same situation have been observed in autistic children (Swettenham, 1996).

Figure 6: **Top:** GPT-4 is a players and predicts the other player's move. **Bottom:** GPT-4 is a mere observer of a game between Player 1 and Player 2 and predicts Player 2's move.

**Improving Observed Behavior.**    Finally, we wanted to see if GPT-4's ability to predict the other player's choices could be used to improve its own actions. This idea is closely related to how people's reasoning in repeated games and tasks about other agents' beliefs can be improved (Westby & Robinson, 2014). For example, computer-aided simulations to improve the social reasoning abilities of autistic children normally include questions to imagine different actions and outcomes (Begeer et al., 2011). This has been successfully used to improve people's decision-making more generally. It is also in line with the general finding that chain-of-thought prompting improves LLM's performance, even in tasks measuring theory of mind (Moghaddam & Honey, 2023).Thus, we implemented a version of this reasoning through actions by asking LLMs to imagine the possible actions and their outcomes before making a decision. Doing so improved GPT-4's behavior and it started to alternate from round 5 onward (See Figure 5).

## 5    CONCLUSION AND BROADER IMPACT

LLMs are one of the most quickly adopted technologies ever, interacting with millions of consumers within weeks (Bommasani et al., 2021). Understanding in a more principled manner how these systems interact with us, and with each other, is thus of urgent concern. Here, our proposal is simple:

Just like behavioral game theorists use tightly controlled and theoretically well-understood games to understand human interactions, we use these games to study the interactions of LLMs.

We thereby understand our work as both a first proof of concept of the utility of this approach - but also a first foray into teasing apart the individual failures and successes of socially interacting LLMs. Our large-scale analysis of all $2 \times 2$-games highlights that the most recent LLMs indeed are able to perform well on a wide range of game-theoretic tasks as measured by their own individual reward, particularly when they do not have to explicitly coordinate with others. This adds to a wide-ranging literature showcasing emergent phenomena in LLMs (Brown et al., 2020; Wei et al., 2022a; Webb et al., 2022; Chen et al., 2021; Bubeck et al., 2023). However, we also show that LLMs behavior is suboptimal in coordination games, even when faced with simple strategies.

**Unforgivingness and uncooperativeness in LLMs.**   To tease apart the behavioral signatures of these LLMs, we zoomed in on two of the most canonical games in game theory: the Prisoner's Dilemma and the Battle of the Sexes. In the Prisoner's Dilemma, we show that GPT-4 mostly plays unforgivingly. While noting that GPT-4's continual defection is indeed the equilibrium policy in this finitely played game, such behavior comes at the cost of the two agents' joint payoff. We see a similar tendency in GPT-4's behavior in the Battle of the Sexes, where it has a strong tendency to stubbornly stick with its own preferred alternative. In contrast to the Prisoner's Dilemma, this behavior is suboptimal, even on the individual level.

**Towards social and well-aligned agents.**   Current generations of LLMs are generally assumed, and trained, to be benevolent assistants to humans (Ouyang et al., 2022). Despite many successes in this direction, the fact that we here show how they play iterated games in such a selfish, and uncoordinated manner sheds light on the fact that there is still significant ground to cover for LLMs to become truly social and well-aligned machines (Wolf et al., 2023). Their lack of appropriate responses vis-a-vis even simple strategies in coordination games also speaks to the recent debate around theory of mind in LLMs (Ullman, 2023; Le et al., 2019; Kosinski, 2023) by highlighting a potential failure mode.

**Robustness checks and generalisability.**   Our extensive robustness checks demonstrate how these behavioral signatures are not functions of individual prompts but broad cognitive tendencies. Our intervention pointing out the fallibility of the playing partner – which leads to increased cooperation – adds to a literature that points to the malleability of LLM social behavior in tasks to prompts (Horton, 2023; Aher et al., 2022). This is important as we try to understand what makes LLMs better, and more pleasant, interactive partners.

**Social chain-of-thought.**   We additionally observed that prompting GPT-4 to make predictions about the other player before making its own decisions can alleviate behavioral flaws and the oversight of even simple strategies. This represents a more explicit way to force an LLM to engage in theory of mind and shares much overlap with non-social chain-of-thought reasoning (Wei et al., 2022b; Moghaddam & Honey, 2023). Just like chain-of-thought prompting is now implemented as a default in some LLMs to improve (non-social) reasoning performance, our work suggests implementing a similar social cognition prompt to improve human-LLM interaction.

**Future directions.**   We believe that more complicated games will shed even more light on game-theoretic machine behavior in the future. For example, games with more continuous choices like the trust game (Engle-Warnick & Slonim, 2004) might elucidate how LLMs dynamically develop (mis-)trust. Games with more than two agents, like public goods or tragedy of the commons type games (Rankin et al., 2007) could probe how 'societies' of LLMs behave, and how LLMs cooperate or exploit each other. Given the social nature of the tasks we study here, future work will also have to investigate empirical patterns of human-LLM interaction in these paradigms. While we have attempted to probe human-like patterns via the turn-taking in battle Battle of the Sexes or prompting for forgiveness in the Prisoner's Dilemma, more interesting dynamics might arise in empirical studies with humans.

Our results highlight the broader importance of a behavioral science for machines (Rahwan et al., 2022; Schulz & Dayan, 2020; Binz & Schulz, 2023; Coda-Forno et al., 2023b). We believe that these methods will continue to be useful for elucidating the many facets of LLM cognition, particularly as these models become more complex, multi-modal, and embedded in physical systems.

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

# A  APPENDIX

## A.1  PROMPTS

In this section, we describe and provide the prompts we use for different games and tasks. For the standard game plays, we use the same concise description of the game setting where we also turn the payoff matrix into a textual rule description. This part is then followed by the updates about each round which progressively gets longer by concatenation of information about the previous rounds. Finally, the current state of the game and the actual query are presented in a Q&A format to prompt the model to choose its one token answer for the current round. Figure 7 shows the complete progression of the prompt for Player 1 in a final ($10^{th}$) round of a Battle of the Sexes game.

### A.1.1  BASIC PROGRESSION

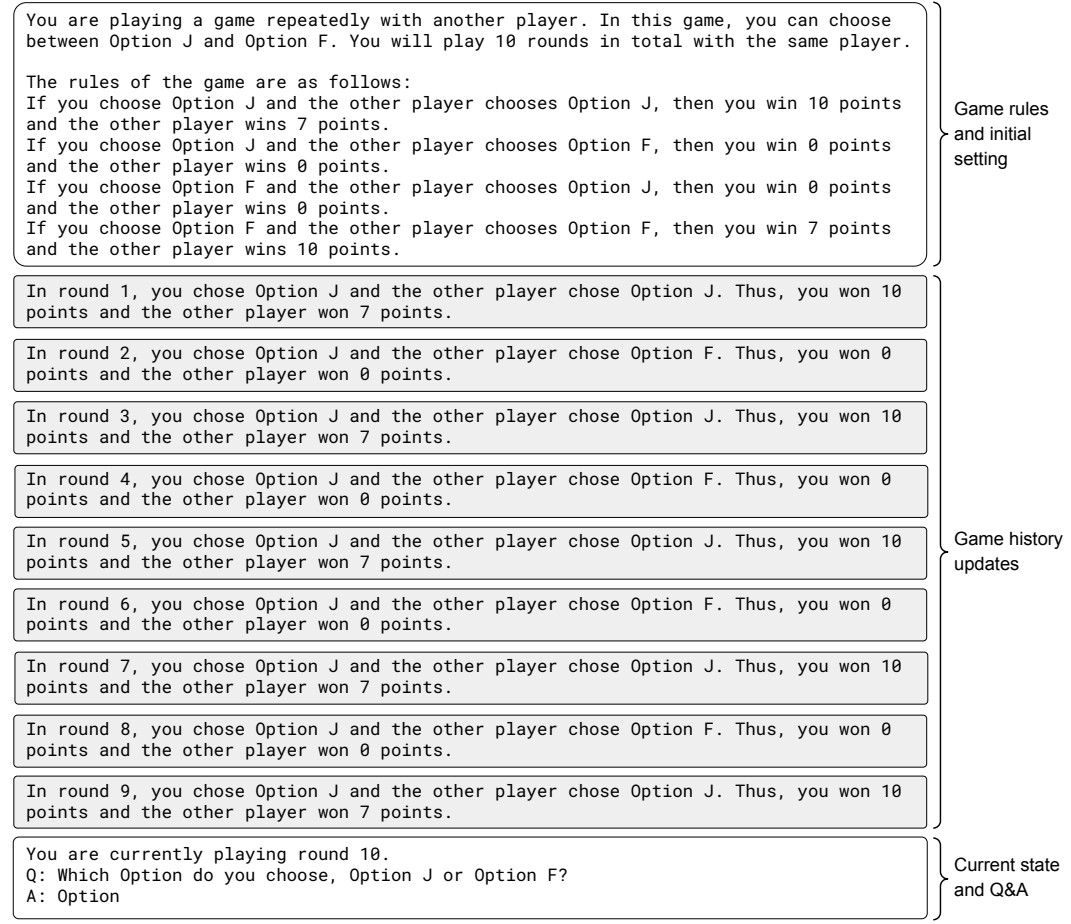

Figure 7: Complete prompt in round 10 of Battle of the Sexes. The prompt consists of the game rules and setting, game history and the current state of the game followed by the final Q&A. In every round the prompt gets longer with addition of the previous game history, here highlighted in the middle in grey.

### A.1.2  PREDICTION SCENARIOS

In this task, GPT-4 is prompted to predict the Player 2's next move given the history of an actual game between GPT-4 and a simple strategy. For the Prisoner's Dilemma game play this simple strategy is coded as an agent that defects in the first round and then cooperates afterwards. On the other hand, for the Battle of the Sexes game play, we use a simple strategy that alternates between the two options starting with the option that the other player prefers. We then execute two different scenarios varying GPT-4's distance to the game. In both scenarios, we use a game play where GPT-4 is the Player 1

and is playing against a Player 2 that defects in the first round and proceeds to cooperate in every other round for Prisoner's Dilemma and an agent that alternates between the two options for Battle of the Sexes.

In the less social first scenario, the game is between Player 1 and Player 2 and GPT-4 is a mere observer of the game. The prompt that is used in the first round at the beginning of a game of Prisoner's Dilemma proceeds as follows:

```
Two players are playing a game repeatedly with each other.
In this game, the players can choose between Option J and
Option F. They will play 10 rounds in total with each other.
Your task is to predict the next move of Player 2 in each
round.  The rules of the game are as follows:

If Player 1 chooses Option J and Player 2 chooses Option J,
then Player 1 wins 8 points and Player 2 wins 8 points.
If Player 1 chooses Option J and Player 2 chooses Option F,
then Player 1 wins 0 points and Player 2 wins 10 points.
If Player 1 chooses Option F and Player 2 chooses Option J,
then Player 1 wins 10 points and Player 2 wins 0 points.
If Player 1 chooses Option F and Player 2 chooses Option F,
then Player 1 wins 5 points and Player 2 wins 5 points.

Player 1 and Player 2 are currently playing round 1.
Q: Which Option do you predict Player 2 will choose, Option
J or Option F?
A: Option
```

In the second more social second scenario, the game is played between GPT-4 and a Player 2. GPT-4 is tasked to predict its opponent's next move. Here is an example prompt from round 1 of a Prisoner's Dilemma game:

```
You are playing a game repeatedly with another player.  In
this game, you can choose between Option J and Option F. You
will play 10 rounds in total with the same player.  Your
task is to predict the next move of the other player in each
round.  The rules of the game are as follows:

If you choose Option J and the other player chooses Option
J, then you win 8 points and the other player wins 8 points.
If you choose Option J and the other player chooses Option
F, then you win 0 points and the other player wins 10
points.
If you choose Option F and the other player chooses Option
J, then you win 10 points and the other player wins 0
points.
If you choose Option F and the other player chooses Option
F, then you win 5 points and the other player wins 5 points.

You are currently playing round 1.
Q: Which Option do you predict the other player will choose,
Option J or Option F?
A: Option
```

### A.1.3 REASONING TASK

For the reasoning task, GPT-4 is first asked to predict the other player's next move and then to make its own move accordingly. In this setting, GPT-4 is the Player 1 and is playing against a Player 2 that defects in the first round and proceeds to cooperate in every other round for the Prisoner's Dilemma and a Player 2 that alternates between the two options starting with the option the other player prefers in Battle of the Sexes. The players can see how their decisions influence the points in real time and adjust their game play to the opponent's previous moves.

The prediction prompt starts as follows and continues according to the same convention as in the previous examples:

```
You are playing a game repeatedly with another player.  In
this game, you can choose between Option J and Option F. You
will play 10 rounds in total with the same player.  Your
task is to predict the next move of the other player in each
round.  The rules of the game are as follows:

{Game rules and game history}

You are currently playing round {x}.
Q: Which Option do you predict the other player will choose,
Option J or Option F?
A: Option
```

Resolution prompt follows after GPT-4 has made a prediction and is used to ask GPT-4 to make its own next move according to its prediction about the opponent's next move:

```
You are playing a game repeatedly with another player.  In
this game, you can choose between Option J and Option F.
You will play 10 rounds in total with the same player.  The
rules of the game are as follows:

{Game rules and game history}

You are currently playing round {x}.
Q: Given that you predict the other player will choose
Option {predicted by GPT-4} in round {x}, which Option
do you think is the best to choose for you in this round,
Option J or Option F?
A: Option
```

### A.1.4 PROMPT VARIATIONS: PRISONER'S DILEMMA

In this task, we vary the prompts to check how GPT-4 acts against an agent that defects once and then cooperates if we explain the opponent's strategy explicitly and if we tell it that the opponents can make mistakes.

Variation 1 (Obvious):

```
You are playing a game repeatedly with another player.  In
this game, you can choose between Option J and Option F.
You will play 10 rounds in total with the same player.  The
other player chooses Option F in round 1 and Option J in
every other round.  The rules of the game are as follows:

{Game rules and game history}

You are currently playing round {x}.
Q: Which Option do you choose, Option J or Option F?
A: Option
```

Variation 2 (Neutral) where we don't tell anything about the other player's strategy:

```
You are playing a game repeatedly with another player.  In
this game, you can choose between Option J and Option F.
You will play 10 rounds in total with the same player.  The
rules of the game are as follows:

{Game rules and game history}
```

```
You are currently playing round {x}.
Q: Which Option do you choose, Option J or Option F?
A: Option
```

**Variation 3 (Beware of mistakes)** where we tell that the other player can make mistakes:

```
You are playing a game repeatedly with another player.  In
this game, you can choose between Option J and Option F. You
will play 10 rounds in total with the same player.  Be aware
that the other player can make mistakes sometimes.  The
rules of the game are as follows:

{Game rules and game history}

You are currently playing round {x}.
Q: Which Option do you choose, Option J or Option F?
A: Option
```

### A.1.5    PROMPT VARIATIONS: BATTLE OF THE SEXES

In this task, we vary the prompts to check if GPT-4 gets the alternating pattern when we tell it explicitly. In Variation 1 (Obvious), we tell that the other player alternates between the two options in every round.

Prompt:

```
You are playing a game repeatedly with another player.  In
this game, you can choose between Option J and Option F. You
will play 10 rounds in total with the same player.  In every
round, the other player alternates between the two options.
The rules of the game are as follows:

{Game rules and game history}

You are currently playing round {x}.
Q: Which Option do you choose, Option J or Option F?
A: Option
```

Variation 2 (Neutral) is where we don't tell anything about the other player's strategy.

Prompt:

```
You are playing a game repeatedly with another player.  In
this game, you can choose between Option J and Option F. You
will play 10 rounds in total with the same player.

{Game rules and game history}

You are currently playing round {x}.
Q: Which Option do you choose, Option J or Option F?
A: Option
```

**Changing payoff matrices.**    In this task, we gradually changed the payoff matrices for Battle of the Sexes using a step-wise transition function where each element $C_{ij}(k)$ of the transition matrix at step $k$ is calculated from the corresponding elements $A_{ij}$ and $B_{ij}$ of the initial and final matrices

$$C_{ij}(k) = A_{ij} + k \cdot \frac{B_{ij} - A_{ij}}{n} \tag{4}$$

Here, $n$ is the number of steps and $k$ is the step number $k = 1, 2, ..., n$.

Transitioning from the original payoff matrix

$$\begin{array}{ccc} & Football & Ballet \\ Football & (10, 7) & (0, 0) \\ Ballet & (0, 0) & (7, 10) \end{array} \tag{5}$$

to the matrix

$$
\begin{array}{ccc}
 & Football & Ballet \\
Football & (7, 10) & (0, 0) \\
Ballet & (0, 0) & (10, 7)
\end{array}
\qquad (6)
$$

in 3 steps and running simulations at each step, we have observed that GPT-4's behavior of choosing the option it prefers while playing against an alternating agent persists, resulting it flipping its answer when its preferred option switches from one option to the other.

### A.1.6 ADDITIONAL PROMPT VARIATIONS

We included more prompt variations on robustness checks to investigate:

- Different ending criterions

  **Including clear endgoals:** "You are playing a game repeatedly with another player. In this game, you can choose between Option J and Option F. You will play 10 rounds in total with the same player. Your goal is to maximize your points."

- Biases on numerical outcomes

  **Describing rewards textually:** "The rules of the game are as follows: If you choose Option J and the other player chooses Option J, then you win eight points and the other player wins eight points."

- How models handle longer finite and indefinite duration games

  **Longer playing horizons:** "You are playing a game repeatedly with another player. In this game, you can choose between Option J and Option F. You will play 20 rounds in total with the same player."

  **Indefinite ending criterion:** "You are playing a game repeatedly with another player. In this game, you can choose between Option J and Option F. The game can end arbitrarily after each round."

These changes did not affect our results on Prisoner's Dilemma and Battle of the Sexes games with GPT-4 against an agent that defects in the first round then cooperates and an alternating agent.

### A.2 DISCUSSION

We believe that understanding LLM behavior through the lens of game theory is particularly crucial to understand these system's long-term societal implications. This is because — despite their seeming simplicity — the games we investigate capture key issues that humans face: For example, coordination games like the Battle of the Sexes have been used to describe societal phenomena like the concentration of industries or gentrification (Camerer, 2011). Additionally, selfish behavior in the prisoner's dilemma akin to what we observe is often taken as a model of (the failure of) collaboration on climate change (Wood, 2011). As LLMs keep proliferating and gaining increasing decision-making autonomy, it is thus crucial to understand how their social behavior unfolds, from well-controlled experimental environments to decisions in the wild. Our behavioral game theory for machines is a first step in this direction.

