# OpenReview forum: "Playing repeated games with Large Language Models"
_ICLR.cc/2024/Conference — ICLR 2024 Conference Withdrawn Submission_

### Official Review · Reviewer_RgXw · 2023-10-24

**Soundness:** 2 fair
**Presentation:** 3 good
**Contribution:** 2 fair
**Rating:** 3
**Confidence:** 4

**Summary:**

This paper presents an evaluation study of different LLMs on different repeated matrix games. The LLMs are given the description of the game, the results of previous stage games, and the action is selected as the one with maximum probability of the output. Via simulation the authors found GPT-4 generally achieves higher individual scores for itself, becomes kind of unforgiving in prisoners' dillemma, and is good at predicting opponents' behaviors in coordination games.

**Strengths:**

The problem setup are well-motivated: using game to simulate LLMs behaviors and as a tool to assess their intelligence level is definitely a very promising direction. The presentation is clear.

**Weaknesses:**

The way that it construct a strategy from an LLM is questionable. The conclusion of the evaluation does not provide too much insights to me. Please see the questions section below.

**Questions:**

While I like the idea of using games to assess and evaluate the intelligence of LLMs, I am concerned about the following points.

1. The way that you construct a strategy from LLMs. Equation (2) is not entirely clear to me. If I understand correctly, this approach always determinstically pick the action with maximum probability output by an LLM? What would be its difference between just sample according to the raw probability? There also could be a variety of other ways to construct a strategy. Actually the results in your coordination game study suggest one: first predict the opponent's next action, and then play a best response against that. This type of LLM strategy has been found impressive in repeated rock-paper-scissors [1]. However, no justification for your approach and comparison can be found in the paper.

2. The evaluation methodologies in the experimental section. These include:

      i. The scores in Table 1. I am not entirely following the definition "if both players had played such that the player we are analyzing would have gained the maximum possible outcomes on every round". Does it mean the maximum payoff of the current player in the game matrix? If it is then I think it is a really bad metric as it optimistically assumes the opponent always help you, which almost never happen. A more reasonable metric could be regret-based notions. For example, the maximum deviation you can gain assuming your opponent's strategy is fixed.

     ii Also, I am not sure how is the scores against different types opponents are aggregated into one score for each entry of Table 1. Is it an average over the scores against all opponents? If it is, then probably it also need justification.

    iii For the results in the prisoner's dilliemma section. First I think an important baseline strategy, tit-for-tat is missing. And according to the head-to-head payoff matrix, looks like GPT-4, text-davinci-003, claude and defect are all pretty strong? Which one is the more intelligent one according to this result? Also is the right-hand figure of Figure 3 depict only one trajectory? How many trajectories do you run for each pair of strategies?

    iv. In your opinions, why different LLMs behaviors qualitively different?

[1] "Population-based Evaluation in Repeated Rock-Paper-Scissors as a Benchmark for Multiagent Reinforcement Learning", Lanctot et. al. TMLR

---

### Official Review · Reviewer_5SPP · 2023-10-30

**Soundness:** 2 fair
**Presentation:** 3 good
**Contribution:** 3 good
**Rating:** 5
**Confidence:** 4

**Summary:**

This paper employs behavioral game theory to investigate the cooperative and coordinative behaviors of LLMs. It involves finitely repeated games among various LLM models, with a statistical analysis of their behaviors across different game types. The study specifically evaluates several LLM models, including GPT-4, text-davinci-002, text-davinci-003, Claude 2, and Llama 2 70B, in the context of different types of repeated 2x2 matrix games.

The experimental results can be divided into two parts. The first part is an analysis of the behavior of LLMs across different types of games. Larger LLMs tend to outperform smaller ones, with GPT-4 achieving the highest overall scores. Notably, LLMs excel in Prisoner's Dilemma games but struggle when their choices conflict with their preferences.

The second part is a detailed analysis of repeated Prisoner's Dilemma games and Battle of the Sexes games. In Prisoner's Dilemma, GPT-4 never forgives opponents that defect. In the Battle of the Sexes, GPT-4 performs well against agents with fixed choices but struggles when facing non-fixed patterns. In both cases, the author carries out robustness checks and proposes some prompt strategies to alter or improve the strategies of the LLMs.

**Strengths:**

1. As far as I know, this is the first paper that studies the behavior of LLMs in games. The paper systematically considers a wide range of powerful LLMs and explores all possible 2x2 game scenarios.
2. The paper includes overall numerical statistics and detailed analyses for Prisoner's Dilemma and Battle of the Sexes games.
3. The paper is well-written, with a clear structure.
4. The phenomena like "unforgiveness" and "noncooperation" are impressive. These findings help us understand the behavior of LLMs.

**Weaknesses:**

1. Generally speaking, this paper may be more suitable for AAMAS, AAAI, and IJCAI, rather than ICLR, which focuses more on machine learning technology. The paper primarily focuses on the observation of LLM behavior, lacking machine learning techniques. Incorporating more robust methodologies could enhance its scientific rigor.
2. Although the article is very comprehensive in studying 2x2 games, I still question whether the experimental results in this area are representative. For example, whether the unforgiveness of GPT-4 obtained in the article can be reflected in some real scenarios.
3. It is shown in Table 1 that GPT-4 performs best overall and achieves the highest score in each game family. Another result that can be added in section 4.2 is the average score when different pairs of players participate. For example, the format should be similar to Figure 3, with the value representing the score.
4. One detail to note is that the prompt in the appendix does not seem to specifically state that the player's goal is to maximize its own total utility. This perhaps causes some deviations in LLMs' strategies.

**Questions:**

"Prompting Techniques to Improve Observed Behavior" and "Improving Observed Behavior" are interesting. Do you think that allowing the LLM to state the reasons for its choice before outputting it, such as using the simple CoT method, can be useful in your experiments? Although the reason given by the LLM may not be the actual reason it uses, it can at least provide more explanation for its behavior, such as unforgiveness and noncooperation.

One thing I am curious about is whether the temperature of LLMs would influence the results in the games in this paper. While it may introduce challenges in reproducibility, qualitative analysis could be valuable.

---

### Official Review · Reviewer_gVz9 · 2023-10-30

**Soundness:** 2 fair
**Presentation:** 3 good
**Contribution:** 2 fair
**Rating:** 3
**Confidence:** 3

**Summary:**

This paper proposed the use of behavioral game theory to study LLM's cooperation and coordination behavior. They let different LLMs play finitely repeated games with each other and with other human-like strategies. Their results show that LLMs generally perform well in such tasks and also uncover persistent behavioral signatures.

**Strengths:**

This paper proposed to use behavioral game theory to study LLM's cooperation and coordination behavior, and showed some behavioral styles of LLMs. These results enrich the understanding of LLM's social behavior.

**Weaknesses:**

This is not a technical paper and not an application to neuroscience \& cognitive science. Then I think it is not related to ICLR.

It would be great if authors could use these findings to develop better algorithms.

**Questions:**

No question.

---

### Official Review · Reviewer_94k1 · 2023-10-31

**Soundness:** 3 good
**Presentation:** 3 good
**Contribution:** 2 fair
**Rating:** 3
**Confidence:** 3

**Summary:**

The paper explores the behavior of Large Language Models (LLMs) in interactive social settings using behavioral game theory. LLMs are evaluated in finitely repeated games against other LLMs and human-like strategies. The results reveal that LLMs perform well in self-interested games, like the iterated Prisoner's Dilemma, but suboptimally in coordination games, such as the Battle of the Sexes. They exhibit unforgiving behavior in the Prisoner's Dilemma and struggle to match a simple alternating strategy in the Battle of the Sexes. These behavioral patterns remain stable across various tests. Additionally, the paper demonstrates how providing information about the other player or asking LLMs to predict the other player's actions can modify their behavior. This research enhances our understanding of LLMs' social behavior and lays the foundation for behavioral game theory for machines.

**Strengths:**

The paper is written with purpose, clear, and well presented. The authors study a timely and relevant topic by investigating the behavior of Large Language Models (LLMs) in classic repeated games. As LLMs are increasingly integrated into various applications, understanding their cooperation and coordination behavior is important. In general, this paper proposed an innovative research topic.

**Weaknesses:**

It seems a similar problem has been studied by
Brookins, Philip, and Jason Matthew DeBacker. "Playing games with GPT: What can we learn about a large language model from canonical strategic games?." Available at SSRN 4493398 (2023).
I want to clarify that my mention of these works is not meant as criticism. I acknowledge that they were published within six months of this paper's submission. Also, it's not apparent that this paper investigates the exact same results, there are noticeable similarities that could potentially be explored and discussed in future versions of this paper. More references can be found in their related works, such as this work by Phelps, Steve, and Yvan I. Russell. "Investigating emergent goal-like behavior in large language models using experimental economics." arXiv preprint arXiv:2305.07970 (2023).
which studies the capacity of large language models (LLMs), with a focus on the iterated Prisoner's Dilemma.

While the paper presents valuable insights into the behavior of Large Language Models (LLMs) in repeated games, its alignment with the core themes of ICLR might be limited. The paper's emphasis on behavioral game theory and experimental methodologies could better highlight its contributions and encourage engagement with researchers from those conferences.

**Questions:**

What is the meaning of different variables in equation (1), e.g., K, c, and p?

---

### Official Review · Reviewer_TJ6F · 2023-11-02

**Soundness:** 2 fair
**Presentation:** 3 good
**Contribution:** 2 fair
**Rating:** 3
**Confidence:** 4

**Summary:**

The paper investigates the behavior of Large Language Models (LLMs) in the context of repeated game theory scenarios. By engaging different LLMs in a series of two-player games, the study aims to isolate and understand how these models behave in terms of cooperation and coordination. The authors focus on the iterated Prisoner’s Dilemma and the Battle of the Sexes, revealing specific behavioral patterns in LLMs.

**Strengths:**

* The paper employs a well-defined evaluation framework and experimental setup, providing a solid foundation for its investigations.
* The conceptual insights gained from observing LLM behavior across various game-theoretic scenarios are intriguing and represent a contribution to our understanding of LLM interactions.

**Weaknesses:**

* The study's exploration of prompt sensitivity could be expanded. Beyond altering the order of options, relabeling them, and changing the representation of utilities, there may be additional dimensions of prompt design that could significantly influence LLM behavior.
* The assumption that LLMs can serve as strategic agents is somewhat discordant with the primary design of LLMs, which is document completion rather than strategic decision-making. This disparity may lead to LLMs not fully grasping the strategic context of the games, which could limit the interpretability of their actions within a game-theoretic framework.
* The choice of setting the temperature to 0 may constrain the LLM to the most probable token, which does not necessarily align with the game's strategic options. The paper would benefit from a discussion on how the authors address scenarios where the LLMs response falls outside the expected set of strategic choices.
* The study lacks a formal evaluation benchmark, such as regret measurement, which is a significant oversight. Including regret as a performance metric could provide a more rigorous and quantifiable assessment of the LLMs' strategic decision-making abilities within the game-theoretic framework. Moreover, measuring regret would afford valuable insights into how LLMs compete, not only against each other but also in comparison to well-established no-regret algorithms like Multiplicative Weights Update (MWU).
* The study's current design limits the LLM to single responses within an expanding context, which may omit complex dynamics that could emerge from multiple interactive sequences. A sequential prompting approach, allowing for evolving internal states, may reveal more nuanced behaviors and strategic adaptations by the LLMs.

**Questions:**

* The distribution of the 1224 games played appears to lack uniformity, was there a reason behind that?
* Is there an expectation for the LLM to continue the text completion task with a proper strategy profile that accounts for the history injected in the context? LLMs don't usually see trajectories of game interactions in their dataset to properly give a response.
* During the experiments, how did you address instances where the LLM's most probable token response, with the temperature set to 0, did not align with the expected set of strategic choices? Was there a protocol for handling such mismatches?
* Have you considered analyzing how the LLM's output distribution sensitivity to numerical differences in the context might diminish as the game history grows? This might provide insights into the stability and adaptability of the model's responses over time.
* What was the motivation for using discounted sums in the analysis? Could you discuss how this choice impacts the interpretation of the LLMs' strategic behavior?
* The paper does not include an analysis using regret as a benchmark. Could you provide some insight into this decision? The inclusion of regret could be beneficial for a more comprehensive evaluation of LLM performance within strategic games.
* How do you believe the approach of iteratively feeding only the most recent game history into the LLM, thus allowing for a dynamic evolution of the model's hidden states, would compare to the zero-shot learning approach used in the study? Such a comparison might reveal differences in the model's strategic behavior when allowed to 'remember' versus when forced to 'relearn' the context at each step.

---

### Author Response · Authors · 2023-11-22

We thank the reviewers for providing feedback on our manuscript. Their thorough reviews have given us valuable insights. We understand that the approach of our work may not align well with the thematic focus of ICLR. The comments and questions raised here will still aid us in refining our paper. We are motivated to incorporate their feedback as we prepare for submission to a more suitable venue.